# Effect of Calcium Ion Supplementation on Oral Microbial Composition and Biofilm Formation In Vitro

**DOI:** 10.3390/microorganisms10091780

**Published:** 2022-09-03

**Authors:** Bhumika Shokeen, Elaine Pham, Julia Esfandi, Takeru Kondo, Hiroko Okawa, Ichiro Nishimura, Renate Lux

**Affiliations:** 1Division of Biosystems and Function, UCLA School of Dentistry, Los Angeles, CA 90095, USA; 2Weintraub Center for Reconstructive Biotechnology, Division of Regenerative & Reconstructive Sciences, UCLA School of Dentistry, Los Angeles, CA 90095, USA; 3Division of Molecular & Regenerative Prosthodontics, Tohoku University Graduate School of Dentistry, Sendai 980-8575, Japan

**Keywords:** oral biofilm, calcium, extracellular DNA, 16S rRNA sequencing, periodontitis

## Abstract

The oral cavity contains a variety of ecological niches with very different environmental conditions that shape biofilm structure and composition. The space between the periodontal tissue and the tooth surface supports a unique anaerobic microenvironment that is bathed in the nutrient-rich gingival crevicular fluid (GCF). During the development of periodontitis, this environment changes and clinical findings reported a sustained level of calcium ion concentration in the GCF collected from the periodontal pockets of periodontitis patients. Here, we report the effect of calcium ion supplementation on human oral microbial biofilm formation and community composition employing an established SHI medium-based in vitro model system. Saliva-derived human microbial biofilms cultured in calcium-supplemented SHI medium (SHICa) exhibited a significant dose-dependent increase in biomass and metabolic activity. The effect of SHICa medium on the microbial community composition was evaluated by 16S rRNA gene sequencing using saliva-derived microbial biofilms from healthy donors and periodontitis subjects. In this study, intracellular microbial genomic DNA (iDNA) and extracellular DNA (eDNA) were analyzed separately at the genus level. Calcium supplementation of SHI medium had a differential impact on iDNA and eDNA in the biofilms derived from healthy individuals compared to those from periodontitis subjects. In particular, the genus-level composition of the eDNA portion was distinct between the different biofilms. This study demonstrated the effect of calcium in a unique microenvironment on oral microbial complex supporting the dynamic transformation and biofilm formation.

## 1. Introduction

The National Health and Nutrition Examination Survey of the U.S. civilian non-institutionalized population reported that 46% of dentate adults (64.7 million people) suffered from periodontitis [1]. The prevalence of periodontitis was positively associated with increasing age and with 8.9% of the people having developed severe or aggressive periodontitis. The periodontal pocket provides a unique anaerobic environment for microbial colonization, and environmental changes in this special oral ecological niche impact the development of dysbiosis in oral commensal microbial biofilms [2]. A recent systematic review reported that the majority of studies found an increased salivary calcium concentration in subjects with chronic periodontitis [3]; thus, it has been postulated that calcium ion contributes to a modified oral environment associated with periodontal diseases [4,5,6].

A major contributor to the nutrient-rich environment in the periodontal pocket is the gingival crevicular fluid (GCF), which is primarily derived from microvascular leakage of gingival tissue and comprises serum electrolytes, proteins, and immune cells, among others [7]. In a classic cohort study of school children, elevated calcium concentrations in the GCF showed a strong correlation with increased gingival inflammation [8]. While sodium and potassium concentrations in GCF are less than 1% of salivary concentrations, the GCF calcium concentration maintains a disproportionately high concentration [9,10,11]. The relatively high concentration of calcium in GCF has been postulated to play a role in subgingival plaque formation [12]. It has been reported that dental plaque contains much higher concentrations of calcium than GCF and that dental plaque samples from subjects with adult periodontitis showed significantly higher calcium concentration than those of healthy subjects [13,14] or those of juvenile periodontitis patients [15].

Taken together, we hypothesized that increased concentrations of calcium ions influence the development of dental plaque and/or its microbial composition, leading to a pathological oral microenvironment. The present limited pilot study addressed the effect of calcium ions on oral microbial biofilm formation in vitro using a previously established complex medium (SHI medium) that allows oral biofilm growth largely reflective of the original inoculum [16,17]. The SHI medium culture system has been used for oral microbial biofilm investigations [18,19] and a study showed that certain modifications allowed a more focused representation of microbial subgingival communities associated with periodontal disease [20]. Here, the original SHI medium was employed to explore the specific effect of calcium on various aspects of oral biofilm formation using saliva pools with mixed oral health status, as well as saliva from individual subjects with a healthy periodontium and those with periodontal disease.

## 2. Materials and Methods

### 2.1. Saliva Collection and Processing

Saliva was collected from periodontally healthy volunteers with gingival sulcus depth of <4 mm and patients diagnosed with periodontitis under UCLA-IRB 13-001075. For saliva collection, volunteers were asked to expectorate 5 mL of saliva into 50 mL sterile falcon tubes. The saliva was then diluted to 25% by adding 15 mL of phosphate-buffered saline (1X PBS) prior to low-speed centrifugation at 2600× *g* for 10 min to pellet large debris and eukaryotic cells. The supernatant was transferred into a new tube containing glycerol at a final concentration of 15%, and 1 mL aliquots were stored in cryotubes at −80 °C. For pooled samples, saliva of volunteers with healthy periodontal status was mixed in equal proportions prior to storage.

In the initial study, salivary samples of six healthy volunteers were mixed at equal volumes to generate one experimental sample, similar to earlier publications [16]. In the subsequent validation study, we collected separate salivary samples from two healthy subjects and two patients with periodontitis. These salivary samples were not pooled and were used as independent experiments.

### 2.2. Oral Microbial Community and Culture Conditions

Human-saliva-derived oral community was grown anaerobically (10% CO_2_, 10% H_2_, and 80% N_2_) at 37 °C according to previously described protocols [16,17]. Briefly, 100 µL of saliva stock derived from a pool of individual healthy or periodontitis patients was inoculated and grown in 1 mL of SHI medium [16] for 17–18 h under anaerobic conditions. Prior to use, this overnight grown oral community was pelleted and washed with 1X PBS. For biofilm seeding, cells were diluted in SHI medium alone or SHI medium supplemented with 1 or 5 mM CaCl_2_ (SHICa) to an optical density of 0.1 at 600 nm. One mL of this diluted oral community was seeded into sterile 24-well polystyrene plates (Fisher Scientific, Hampton, NH, USA) and incubated under anaerobic conditions at 37 °C for up to sevendays. The biofilm communities were assessed after one (D1), three (D3), five (D5) and seven (D7) days. For all experiments, the medium was carefully removed from the 24-well polystyrene plates at the end of the respective incubation period and washed one time with 1 mL sterile 1X PBS prior to further processing.

### 2.3. Crystal Violet Assay

The total biomass of the biofilms accumulated on the polystyrene plates was assessed using the crystal violet assay as described earlier [21]. Briefly, the PBS-washed wells of the 24-well polystyrene plate were dried prior to fixation of the biofilms with 1 mL methanol. After 15 min, methanol was carefully removed, and the biofilms were submerged in 1 mL of 0.5% crystal violet aqueous solution for 15 min. The wells were then washed four times with distilled water to remove excess crystal violet and ensure complete removal of residual dye. After the final wash, 1 mL of 33% acetic acid was added to the wells and mixed to extract the crystal violet. The acetic acid solution containing the crystal violet stain retained by the biofilms was transferred into 1.5 mL cuvettes (USA Scientific, Inc., Ocala, FL, USA) and the optical density at 570 nm was determined. All experiments were performed as biological and technical triplicates.

### 2.4. MTT Assay

A 3-(4,5-dimethylthiazol-2-yl)-2,5-diphenyl tetrazolium bromide (MTT; Sigma-Aldrich, St. Louis, MO, USA) solution was aseptically prepared by dissolving the MTT powder at a concentration of 5 mg/mL in sterile 1X PBS at room temperature. The solution was stored at 4 °C in a dark, screw-cap container. After washing the biofilms with 1X PBS, the biofilm cells were removed from the well by scraping and resuspended under vigorous pipetting into 1 mL of 1X PBS. An aliquot of 100 µL for each of the biofilm suspensions was placed into a 96-well plate and 10 µL of MTT solution was added to each well. The 96-well plate was then incubated for 2 h at 37 °C at 120 rpm in a shaker. After the incubation period, an equal amount of dimethyl sulfoxide (100 µL) was added to each well to solubilize formazan, the insoluble purple cleavage product of MTT generated through the metabolism of live cells. The levels of microbial metabolic activity in the biofilms were determined by measuring the absorbance at OD_570_ nm.

### 2.5. DNA Extraction

After the individual incubation periods, planktonic cells were removed from the biofilms by gently aspirating the medium and washing once with 500 µL of 1X PBS. Following the wash, 250 uL of 1X PBS was added into each well and microbial cells were harvested by scraping with a sterile pipette tip and mixing well with the pipette. After combining the cells harvested from three wells for each condition, the bacterial-biofilm-derived cells were further pelleted at 3250× *g* for 15 min at 4 °C. While the bacterial pellet was used for intracellular DNA (iDNA) isolation, the supernatant was further processed for isolation of extracellular DNA (eDNA). Prior to isolation of eDNA, the supernatant was filtered through 0.22 µm syringe filters to remove residual bacterial cells.

*iDNA extraction*: The bacterial pellet was used for iDNA extraction with the Epicentre MasterPure DNA Extraction and Purification Kit (Madison, WI) according to manufacturer’s instructions, with some minor modifications [22].

*eDNA extraction*: eDNA was extracted from the cell-free supernatants of the microbial biofilm according to the protocol by Liao et al. (2014) with minor modifications. Briefly, the cell-free supernatants (containing eDNA) were mixed with 2 volumes of absolute ethanol and a 1/10th volume of sodium acetate (3 M, pH 5.5). After overnight precipitation at −80 °C, eDNA was pelleted by centrifugation at 13,000 rpm at 4 °C for 20 min followed by a wash with ice-cold 70% ethanol. The eDNA was then air-dried and the pellet was resuspended in sterile deionized water. The DNA concentrations of the iDNA and eDNA samples were quantified using Nanodrop.

### 2.6. PCR and Denaturing Gradient Gel Electrophoresis (DGGE)

Both iDNA and eDNA were amplified using the universal 16S primers Bac1 with a GC clamp (5′-CGC CCG CCG CGC CCC GCG CCC GTC CCG CCG CCC CCG CCC GAC TAC GTG CCA GCA GCC-3′) and Bac2 (5′-GGA CTA CCA GGG TAT CTA ATC C-3′) to amplify a 300 bp region of the 16S ribosomal RNA gene as described (Rupf et al. 1999 [23]). The PCR was performed in a total reaction volume of 50 µL containing 25 µL of 2X GoTaq^®^ Green Master Mix (Promega, Madison, WI, USA), 0.5 μM each of forward and reverse primers, and 50 ng of DNA. The PCR conditions consisted of an initial denaturation for 3 min at 95 °C followed by 35 cycles of denaturation at 95 °C for 1 min, annealing at 56 °C for 1 min, and elongation at 72 °C for 30 s. After 35 cycles of amplification, an additional elongation step was performed at 72 °C for 5 min. Before separating the PCR products via DGGE (denaturing gradient gel electrophoresis), PCR amplification was confirmed by gel electrophoresis on a 1.0% agarose gel. The DGGE was performed using the Bio-Rad DCode™ Universal Mutation Detection System (Bio-Rad Laboratories, Inc., Hercules, CA, USA). Gradient polyacrylamide gels ranging from 40% to 60% denaturants (urea and formamide) were prepared. Approximately 45 μL of PCR product was loaded into each well for separation by electrophoresis at 60 V for 17–18 h at 58 °C. After electrophoresis, the gels were stained with SYBR safe and images were taken with a Molecular Imager Gel Documentation system (Bio-Rad Laboratories).

### 2.7. 16S rRNA Gene Sequencing and Data Analysis

The 16S rRNA gene sequencing was performed at Laragen Sequencing & Genotyping (Culver City, CA, USA) with minor modifications of the Illumina 16S metagenomics library preparation protocol. Briefly, BAC357TS-for (CCTACGGGNGGCWGCAG) and BAC806TS-rev (GACTACHVGGGTATCTAATCC) primers with Illumina overhang adapters were used to amplify the V3–V4 regions of the 16S rRNA. The amplicons were purified with AMPure XP beads (Beckman Coulter, A63881, Brea, CA, USA) and indexed using the Nextera XT Index kit (Illumina, San Diego, CA, USA). Before quantification with the DNA KAPPA kit (ROCHE Diagnostics, Basel, Switzerland), the amplicons were again purified with AMPure XP beads (Beckman Coulter, A63881). Equal amounts of each sample were then pooled into a single library. The quantity and quality of the library was checked at Laragen Sequencing & Genotyping (Culver City, CA, USA) before paired end sequencing (2 × 300 bp) on the Illumina Miseq platform.

Demultiplexed sequences were obtained from Laragen Sequencing & Genotyping (Culver City, CA, USA) and imported into QIIME2 (v2020.11, [24]. Low-quality sequences containing bases with Phred quality values of <20 were trimmed and denoised using the DADA2 package [25]. As the quality of reverse reads was low, the amplicon sequence variants (ASVs) generated from forward reads were taxonomically assigned by comparison to the HOMD database [26]. Alpha and beta diversity analyses were performed using the core metrics plugin in QIIME 2. The Shannon’s index diversity measure was used for calculating alpha diversity, while beta diversity was assessed by weighted unifrac. For calculations, the data were rarefied to 39,159 sampling depth, the lowest number of sequences found amongst all samples.

### 2.8. Statistical Analysis

All statistical analyses were performed using GraphPad Prism 9.1.0 (GraphPad Software, Inc, San Diego, CA, USA). The significance of alpha diversity measures was assessed by the Kruskal–Wallis test, and beta diversity measures were evaluated using analysis of similarity (ANOSIM) with 999 permutations in QIIME2 [24].

## 3. Results

### 3.1. Biofilm Biomass and DGGE Profile with CaCl_2_ Supplementation

Healthy-human-saliva-derived biofilms grown in SHI medium in the presence and absence of 1 mM CaCl_2_ and 5 mM CaCl_2_ demonstrated a significant effect of CaCl_2_ supplementation by the observed robust increase of microbial biomass (Figure 1A). In particular, 5 mM CaCl_2_ supplementation nearly doubled the microbial biomass revealed by crystal violet-staining compared to control SHI medium at D3 and reached a plateau. The supplementation of 1 mM CaCl_2_ showed a delayed effect in biomass increase and reached a plateau at D5.

Saliva has been reported to contain 0.6~2 mM calcium ions [27,28] and the subgingival dental plaque-released calcium concentration was shown to be 2~5 mM [29]. In this experiment, 1 mM CaCl_2_ supplementation was used to reflect a normal saliva environment, while 5 mM CaCl_2_ supplementation was used to represent a subgingival dental plaque environment in a periodontal pocket.

Biofilms consist of microbial communities embedded in extracellular matrix. To further investigate the influence of CaCl_2_ on biofilm formation, microbial intracellular DNA (iDNA) was isolated for evaluation of microbial community profiles for the 16S rRNA PCR DGGE profiling. During D1 and D3, DGGE profiles were nearly identical in all groups. However, at D5 and D7, CaCl_2_-supplemented groups showed deviation from the control SHI medium group (Figure 1B).

### 3.2. Microbial Viability with CaCl_2_ Supplementation

The initial investigation revealed a significant effect of CaCl_2_ supplementation on both biomass production and 16S rRNA PCR DGGE microbial community profiles. The differential effect appeared to reach a plateau at D5, which was sustained at D7. Thus, we next investigated the microbial viability in this new in vitro model during the early biofilm formation period from D1 to D5.

As the biofilms matured, the metabolic activity of the microorganisms in the biofilm significantly decreased from D1 to D5 when grown in SHI medium, whereas the SHI medium with 1 mM CaCl_2_ condition maintained the metabolic activity throughout D1 to D5. (Figure 2A). Furthermore, the SHI medium supplemented with 5 mM CaCl_2_ increased microbial metabolic activity from D1 to D3 and from D1 to D5, which were both significant (Figure 2A).

Comparison of metabolic activities under different medium conditions on the same day revealed that on D1, SHI exhibited the highest values and SHI medium +5 mM CaCl_2_ the lowest, with the difference being significant. This trend was reversed as the biofilm matured on D3 and D5. On D3, there was a significant increase in metabolic activity of biofilm cells in SHI medium + 5 mM CaCl_2_ in comparison to both SHI medium and SHI medium + 1 mM CaCl_2_, and this trend was sustained_._ on D5.

### 3.3. eDNA Profile

The results presented here indicate that increased calcium levels significantly affected biofilm biomass but also metabolic activity and viability. It is a well-known fact that in most biofilms, microorganisms only constitute as little as 10% of the dry mass, while the rest is composed of the different extracellular matrix (ECM) components [30,31]. Among the ECM components, eDNA plays a pivotal role in various stages of biofilm formation ranging from bacterial adhesion, aggregation, to determination of biofilm architecture [32,33,34,35,36].

We examined the 16S rRNA PCR DGGE profile of eDNA (Figure 2B), which was different from the corresponding iDNA profiles (Figure 1B). Interestingly, on D3, the eDNA released by the community grown in SHI + 5 mM CaCl_2_ showed a distinct profile, with a marked increase in GC-rich taxa that migrate faster through the DGGE. This difference was further amplified on D5 when eDNA showed distinct profiles between the various medium conditions (Figure 2B).

### 3.4. Biomass and Viability of Biofilms from Saliva of Healthy and Periodontitis Patients

In the experiments using biofilms grown from a pool of saliva from healthy donors, the highest biofilm biomass was observed on D5 in the presence of 5 mM CaCl_2_. Thus, to further investigate the behavior of biofilm derived from the saliva of healthy subjects or periodontitis patients, biofilm biomass and metabolic activity was assessed in SHI medium and SHI medium + 5 mM CaCl_2_. Validating the early observation of significantly elevated biomass and metabolic activity in the presence of CaCl_2_, a significant increase was observed between SHI medium and SHI medium + 5 mM CaCl_2_ for biofilms derived from the saliva of both healthy and periodontitis patients (Figure 3A,B).

### 3.5. 16S rRNA Sequencing of iDNA

To elucidate the microbial composition of saliva-derived biofilms of healthy and periodontitis patients, iDNA obtained from biofilms grown under both medium conditions was subjected to 16S rRNA sequencing.

The alpha diversity analysis by Shannon’s index (Figure 4A) revealed a decrease in alpha diversity for iDNA in SHI + 5 mM CaCl_2_ biofilms that were seeded with the saliva from healthy individuals. Biofilms derived from the saliva of periodontitis patients, in contrast, exhibited an increase in alpha diversity in SHI + 5 mM CaCl_2_.

Except for minor differences in the abundance of some genera, the iDNA of both healthy subjects and periodontitis patients showed similar community profiles in SHI medium and SHI medium + 5 mM CaCl_2_ (Figure 4B).

### 3.6. Beta Diversity of iDNA and eDNA

The beta diversity analysis with weighted unifrac analysis (Figure 5A) revealed a significant difference between the microbial communities represented by iDNA and their contribution to eDNA. Independent of the original sample type (saliva from individuals with healthy periodontium or those with periodontal disease) or growth conditions (SHI or SHI + 5 mM CaCl_2_), iDNA clustered closely together (Figure 5B). Interestingly, the eDNA released from the biofilms grown in SHI medium + 5 mM CaCl_2_ from the saliva of individuals with periodontitis were distinct, while the eDNA isolated from the biofilms formed under all other medium conditions were very similar to each other.

### 3.7. Microbial Contribution to eDNA

The microbial contribution to eDNA was evaluated by 16S rRNA sequencing. The alpha diversity of eDNA of healthy subjects showed only a slight decrease in SHI medium + 5 mM CaCl_2._ In contrast, eDNA of periodontitis patients exhibited striking differences (Figure 6A).

Genus-level microbial composition of the eDNA further demonstrated that microbial contribution to eDNA in healthy subjects was not affected by 5 mM CaCl_2_ supplementation. However, 5 mM CaCl_2_ supplementation strongly modified genus-level microbial composition that contributed to eDNA of periodontitis patients (Figure 6B). The effect of 5 mM CaCl_2_ supplementation on microbial contribution to eDNA of periodontitis patients showed striking contrast to the iDNA behavior (Figure 5B).

While many genera exhibited apparent differences in relative abundance, these were significant only for *Peptostreptococcus* and *Veillonella* for eDNA of biofilms derived from the saliva of periodontitis patients in SHI + 5 mM CaCl_2_ when compared to its iDNA counterpart or the corresponding eDNA from biofilm grown from the saliva of patients without periodontal disease (Figure 7). Specifically, the relative abundance of *Peptostreptococcus* decreased and the relative abundance of *Veillonella* increased in the presence of CaCl_2_.

## 4. Discussion

Calcium has emerged as an important structural component for biofilm formation that supports intricate 3D structures and contributes to biofilm fitness [37,38,39,40]. Calcium binds to eDNA and exopolysaccharides (EPS) in the biofilm extracellular matrix (ECM) [41,42] and is thought to play an integral role in establishing the sophisticated biofilm macrostructure of channels and diffusion barriers that enable transport of nutrients into the biofilm and keep out harmful substances [40]. A study using a single cell culture of *Pseudomonas fluorescens* with 1.5 mM and 15 mM calcium supplementation highlighted the increased adhesive nature of the biofilms, and increased production of EPS [43].

Studies exploring the role of calcium in oral biofilms are focused on caries-related processes and supragingival monospecies biofilm models [44,45]. In this study, we explored the effect of calcium on biofilm formation, metabolic activity, and viability, as well as the microbial composition of biofilm communities and the corresponding eDNA in the context of calcium concentrations relevant in periodontal diseases using an established in vitro model system reflective of complex oral microbial biofilms [16,17]. Increased calcium levels have been regarded as one of the hallmarks of periodontal disease [6,14,16,17], and the presence of large calcium reservoirs in plaque [46] has been found to be involved in maintaining the plaque structure [47,48,49]. Our results indicate that calcium enhanced biofilm biomass and increased metabolic activity and viability of oral microorganisms in a concentration-dependent manner (Figure 1 and Figure 2). Further investigation of the of the microbial community composition included, for the first time, isolation and characterization of eDNA and iDNA from complex in vitro oral biofilms.

Previous reports have implicated calcium ion levels in biofilm formation, stability, and increased cohesiveness [50,51]. In the present study, an increased biomass was observed in the presence of calcium in the biofilms derived from pooled saliva (Figure 1A) as well as individual saliva samples from healthy subjects and periodontitis patients (Figure 3A). The biomass accumulation in the presence of calcium has been reported using single species microbial culture for *Streptococcus mutans* [52], *Pseudomonas aeruginosa* [51], or *Bacillus subtilis* [53], as well as a defined anaerobic multispecies biofilm [47].

The novel observation in the present study was the unexpectedly large effect of calcium on microbial community contributing to eDNA in biofilms (Figure 2B and Figure 6). A total of 5 mM calcium supplementation triggered a distinct eDNA release in oral microbial communities grown from periodontitis patient samples but not the one derived from patients with a healthy periodontium (Figure 6). This finding is intriguing as it implies the possibility that the species present in the biofilms derived from periodontitis patients respond differently to the disease trigger calcium despite their genus-level similarities with the other biofilms examined in this study.

Calcium enhances eDNA release in *S. mutans* [54], plays a pivotal role in stabilizing the biofilms by forming calcium ion bridges between eDNA molecules [41], and provides structural integrity to bacterial cell envelopes [55,56,57]. Bacterial cells and EPS are usually negatively charged, and cations such as calcium form bridges between them, stabilizing the biofilm in the process. eDNA is released actively by bacterial vesicles [58,59,60], lysis of bacterial cells [33,35,61], or physical and chemical environmental mediators [62,63]. While the majority of past studies investigating eDNA were conducted using microbial biofilm derived from soil samples [64,65], more recently the role of eDNA has been increasingly characterized in pathological biofilms within their biological contexts [66,67,68]. The pilot in vitro study presented here characterized eDNA along with iDNA and provides, to the best of our knowledge, the first report of eDNA isolation and identification of the genera represented in the eDNA portion of oral biofilms. As evidenced by the DGGE and 16S rRNA sequencing results, the composition of eDNA was quite distinct from the corresponding iDNA of the microbial biofilm community (Figure 2B and Figure 6). Interestingly, addition of calcium impacted eDNA composition, while the iDNA community profile remained mostly unaffected, consistent with a previously reported role for calcium in eDNA release [54].

Detailed genus-level comparison of iDNA and eDNA (Figure 6) further confirmed overall similarities between the iDNA microbial profiles of both healthy- and periodontitis-patient-saliva-derived biofilms. In contrast, the composition of eDNA was strikingly different from the respective iDNA of saliva-derived biofilms. Consistent with beta diversity findings, the eDNA of saliva-derived biofilms from periodontitis patients was most prominently impacted by calcium supplementation. While several genera exhibited apparent differences between the conditions tested, only the decrease of *Peptostreptococcus* and the increase in *Veillonella* in the eDNA derived from calcium-supplemented biofilm grown from the saliva of periodontitis patients were significant in comparison to the corresponding unsupplemented biofilms or the ones from periodontally healthy individuals (Figure 7). This suggests that presence of calcium may lead to changes in eDNA, either through lysis of bacteria or through active release of eDNA in periodontitis.

## 5. Conclusions

In summary, this pilot study of the effect of calcium on oral complex biofilm communities indicated that elevated calcium levels in the periodontal pocket during disease onset not only enhanced biofilm formation but could also serve as a trigger for differential eDNA release that may further contribute to disease development. Future studies should include larger samples size and use subgingival plaque samples rather than salivary inocula. Additionally, the newly adapted version of SHI medium that supports disease-associated subgingival microbiota could be used in combination with metagenomic sequencing to capture species-level biofilm community composition and gain a more complete picture of the nature of the eDNA released.

## Figures and Tables

**Figure 1 microorganisms-10-01780-f001:**
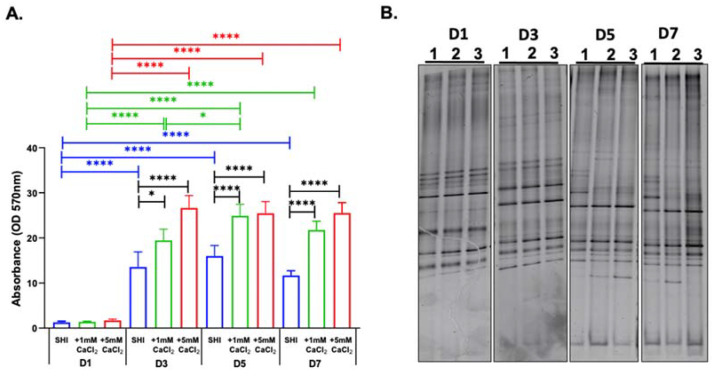
Effect of calcium ion supplementation on human-saliva-derived oral biofilms. The biofilms were grown in SHI medium (SHI), SHI medium + 1 mM CaCl_2_ (+1 mM CaCl_2_), and SHI medium + 5 mM CaCl_2_ (+5 mM CaCl_2_), and evaluated at day 1 (D1), day 3 (D3), day 5 (D5), and day 7 (D7). (**A**) Biofilm biomass: The biomass was assessed by measuring crystal violet at the optical density of 570 nm. Black bars indicate comparisons between conditions at the same time point, whereas comparisons of the same medium condition between time points are indicated in blue (SHI), green (+1 mM CaCl_2_), and red (+5 mM CaCl_2_). * Represents *p* ≤ 0.05 and **** represents *p* ≤ 0.0001. (**B**) Microbial composition: Denaturing gradient gel electrophoresis (DGGE) profiles of iDNA (intracellular DNA) of oral microbial biofilms grown in 1. SHI medium, 2. SHI medium + 1 mM CaCl_2_, and 3. SHI medium + 5 mM CaCl_2_.

**Figure 2 microorganisms-10-01780-f002:**
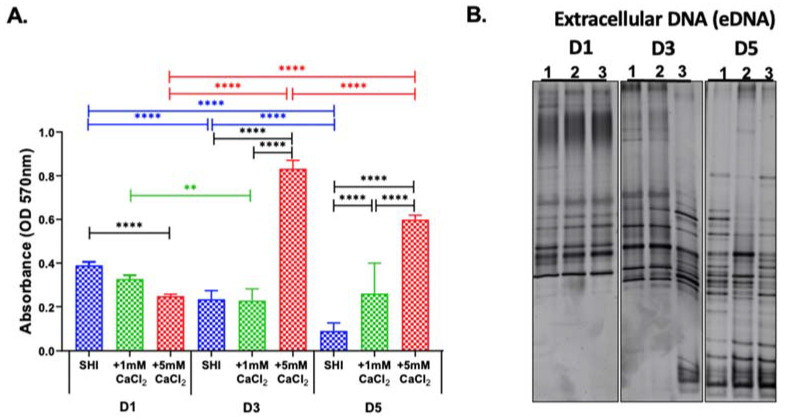
Characterization of oral microbial biofilms grown in SHI medium with CaCl_2_ supplementation evaluated at day 1 (D1), day 3 (D3), and day 5 (D5). (**A**) Biofilm metabolic activity: The metabolic activity of the biofilms was assessed by measuring formazan formation from MTT at the optical density of 570 nm. ** Represents *p* ≤ 0.01, and **** represents *p* ≤ 0.0001 (**B**) Extracellular DNA (eDNA) composition: eDNA is a ubiquitous and pivotal component of biofilm. Denaturing gradient gel electrophoresis (DGGE) profiles of eDNA of oral microbial biofilms grown in 1. SHI medium, 2. SHI medium + 1 mM CaCl_2_, and 3. SHI medium + 5 mM CaCl_2_ at day 1 (D1), day 3 (D3), and day 5 (D5).

**Figure 3 microorganisms-10-01780-f003:**
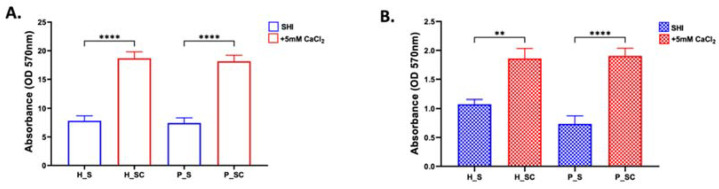
Effect of calcium ion supplementation on biofilms derived from the saliva of periodontally healthy (H) and periodontitis (P) patients. The biofilms were grown in SHI medium (blue bars) and SHI medium + 5 mM CaCl_2_ (red bars) for five days. The bar plots representing the mean and standard deviation of three independent experiments performed in triplicate are shown. (**A**) Biofilm biomass: as assessed by measuring crystal violet at the optical density of 570 nm. (**B**) Biofilm metabolic activity: measured by optical density of formazan formation from MTT at 570 nm. ** Represents *p* ≤ 0.01, and **** represents *p* ≤ 0.0001.

**Figure 4 microorganisms-10-01780-f004:**
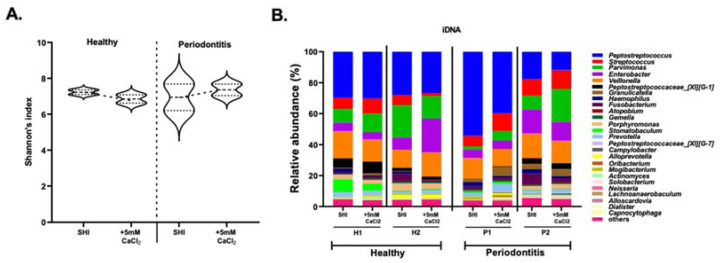
The effect of CaCl_2_ supplementation on iDNA. (**A**) Alpha diversity analysis of iDNA representation of biofilms derived from the saliva of healthy (H) and periodontitis (P) patients. The biofilms were grown in SHI medium (SHI) and SHI medium + 5 mM CaCl_2_ (+5 mM CaCl_2_). Violin plots representing the alpha diversity of iDNA in the absence and presence of +5 mM CaCl_2_ are shown. The median line represents the average while the dotted lines represent minimum to maximum value. Alpha diversity is measured by Shannon’s index, representing mean species richness. (**B**) Genus-level microbial composition of iDNA from healthy (H1, H2) and periodontitis (P1, P2) patients as revealed by 16S rRNA gene sequencing. Bar plots represent the relative abundance of genera present in the oral biofilm communities formed in SHI medium (SHI) and SHI medium + 5 mM CaCl_2_ (+5 mM CaCl_2_).

**Figure 5 microorganisms-10-01780-f005:**
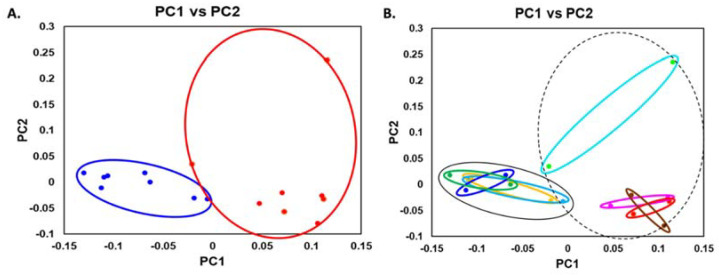
Beta diversity analysis of iDNA and eDNA representation of biofilms derived from the saliva of periodontally healthy (H) and periodontitis (P) patients as evaluated by weighted unifrac. Principal coordinate analysis (PCoA) is plotted according to (**A**) the type of DNA: iDNA (blue circles) and eDNA (red circles), and (**B**) periodontal status of the patients, medium condition (S—SHI; SC—SHI + 5 mM CaCl_2_) and type of DNA: H-S-iDNA (dark blue circles), H-SC-iDNA (dark green circles), P-S-iDNA (light blue circles), P-SC-iDNA (yellow circles), H-S-eDNA (red circles), H-SC-eDNA (pink circles), P-S-eDNA (brown circles), and P-SC-iDNA (turquoise circles). All iDNA samples are indicated by a black solid line; all eDNA samples are indicated by a black dotted line.

**Figure 6 microorganisms-10-01780-f006:**
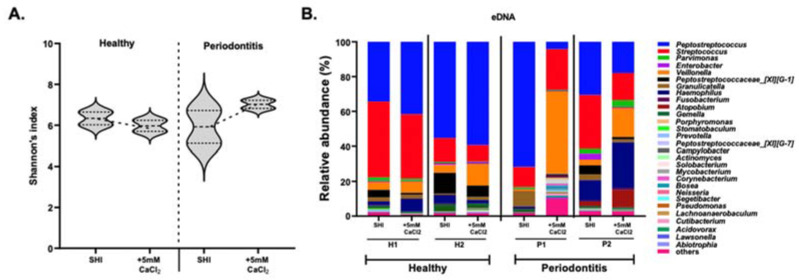
The effect of CaCl_2_ supplementation on eDNA. (**A**) Alpha diversity analysis of eDNA representation of biofilms derived from the saliva of healthy (H) and periodontitis (P) patients. The biofilms were grown in SHI medium (SHI) and SHI medium + 5 mM CaCl_2_ (+5 mM CaCl_2_). Violin plots representing the alpha diversity of eDNA in the absence and presence of +5 mM CaCl_2_ are shown. The median line represents the average while the dotted lines represent minimum to maximum (**B**) Genus-level microbial composition of eDNA in the healthy (H1, H2) and periodontitis (P1, P2) patients as revealed by 16S rRNA gene sequencing. Bar plots represent the relative abundance of genera present in the oral biofilm communities formed in SHI medium (SHI) and SHI medium +5 mM CaCl_2_ (+5 mM CaCl_2_).

**Figure 7 microorganisms-10-01780-f007:**
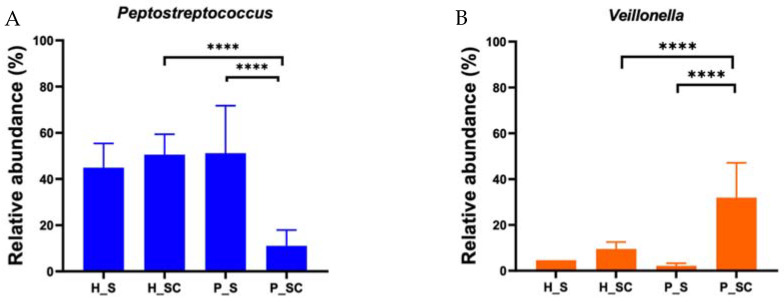
Relative abundance of (**A**) *Peptostreptococcus* (blue bar plots) and (**B**) *Veillonella* (orange bar plots) in eDNA. The bar plots represent the mean of relative abundance of the respective genus in the biofilms derived from two periodontally healthy and two periodontitis patients in both SHI (_S) and SHI + 5 mM CaCl_2_ (_SC). Error bars represent standard error of mean (SEM). **** Represents *p* ≤ 0.0001.

## Data Availability

Data are contained within the article. The 16S rRNA sequencing data will be available at the Sequence Read Archive (SRA).

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
