# Peer review of "Effect of Calcium Ion Supplementation on Oral Microbial Composition and Biofilm Formation In Vitro"

_microorganisms, 2022, doi:10.3390/microorganisms10091780_

Round 1
Reviewer 1 Report
1) In general, the writing seems scientifically immature. The manuscript could use some better phraseology.
2) There is no novelty in the first part of this research which represents the effect of Ca ions on biofilm formation and metabolic activity. This section needs to be rewrite and discuss the results with more details.
3) The results section just focuses on the findings rather than the reason behind that. It needs to be addressed with more details. For example, why there is an increase in metabolic activity or biofilm biomass with increase in Ca concentration. Could it be related to the adhesion kinetics, surface charges, cation bridging interaction and physio-chemical interactions of ions or etc...? What is the role of EPS here?
Safari A, Habimana O, Allen A, Casey E. 2014. The significance of calcium ions on Pseudomonas fluorescens biofilms–a structural and mechanical study. Biofouling. 30:859–869.
Zhang Y, Li C, Wu Y, Zhang Y, Zhou Z, Cao B. 2019. A microfluidic gradient mixer‐flow chamber as a new tool to study biofilm development under defined solute gradients. Biotechnol Bioeng. 116:54–64
4) The author should bold the specific novelties of their findings with other reports. If their novelty is just the extracellular DNA results, they need larger sample size at different time point (add data for D7).
5) The authors didn’t mention how they selected the Ca concentration (why 1 and 5 mM? Please add some references). It would be great if they could have added higher concentration of Ca (10 mM and/or 50 mM) to their results. It has been reported that too much Calcium has a negative effect on bacterial attachment and biofilm forming.
Chen G, Walker SL. 2007. Role of solution chemistry and ion valence on the adhesion kinetics of groundwater and marine bacteria. Langmuir. 23:7162–7169.
Venegas SC, Palacios JM, Apella MC, Morando PJ, Blesa MA. 2006. Calcium modulates interactions between bacteria and hydroxyapatite. J Dent Res. 85:1124–1128.
Mangwani N, Shukla SK, Rao TS, Das S. 2014. Calcium-mediated modulation of Pseudomonas mendocina NR802 biofilm influences the phenanthrene degradation. Colloids Surf B Biointerfaces. 114:301–309
6) It would be great if authors could add some more references for discussion part line 370-410.
7) The authors make many statements regarding that their results are in agreement with previous studies. These statements need to be revised. It has affected the novelty and uniqueness of their work. They should add some scientific arguments that support their claims.
8) It would be great if authors could go through the article, and cover the typos, abbreviations, spacing error, and extra. The format should be constant all over the manuscript: Ex. 15ml or 15 ml (spacing error), day 1 or D1, 2 hrs or 2 hours (the format is not consistent), ul or µl, CaCl2 needs to be fixed (use subscript, CaCl2).
9) Figures’ legend has different size and formatting. This needs to be fixed.
Author Response
In general, the writing seems scientifically immature. The manuscript could use some better phraseology.
RESPONSE: We have reviewed the manuscript accordingly and made the necessary modifications.
There is no novelty in the first part of this research which represents the effect of Ca ions on biofilm formation and metabolic activity. This section needs to be rewrite and discuss the results with more details.
RESPONSE: Figure 1 was revised and now contains an extended observation period of biofilm formation to D7. We found no significant difference between D5 and D7 biofilm formation. Furthermore, we described the time-course change in intracellular DNA 16S rRNA PCR-derived DGGE gene profile from D1 to D7. There was a progressive change in microbial profile until D5. This new information demonstrates the novelty of this in vitro culture system.
The results section just focuses on the findings rather than the reason behind that. It needs to be addressed with more details. For example, why there is an increase in metabolic activity or biofilm biomass with increase in Ca concentration. Could it be related to the adhesion kinetics, surface charges, cation bridging interaction and physio-chemical interactions of ions or etc...? What is the role of EPS here?
Safari A, Habimana O, Allen A, Casey E. 2014. The significance of calcium ions on Pseudomonas fluorescens biofilms–a structural and mechanical study. Biofouling. 30:859–869.
This reference was included in the discussion section. “This study used a single cell culture of Pseudomonas. fluorescens with 1.5 mM and 15 mM Ca2 supplementation. The data highlighted the increased adhesive nature of the biofilm and increased production of EPS”.
Zhang Y, Li C, Wu Y, Zhang Y, Zhou Z, Cao B. 2019. A microfluidic gradient mixer‐flow chamber as a new tool to study biofilm development under defined solute gradients. Biotechnol Bioeng. 116:54–64
This study investigated the effect of BSA, CaCl2 and KNO3 in the solute gradients on biofilm formation using Shewanella oneidensis and Comamonas testosteroni. The biovolume was negatively affected by increasing Ca concentration gradient, while surface roughness of biofilm was decreased at threshold of 0.5 mM. The Calcium concentrations used in this study are lower than the concentrations used in our study. Thus this reference was not relevant to our study.
The author should bold the specific novelties of their findings with other reports. If their novelty is just the extracellular DNA results, they need larger sample size at different time point (add data for D7).
RESPONSE: We appreciate this comment. The revised manuscript contains new information of crystal violet-stained biomass and microbial composition profiles using iDNA 16S rRNA PCR DGGE of D1, D3, D5 and D7. We found that microbial biomass progressively increased under Ca supplementation up to D5 and then plateaued to D7. Similarly, microbial composition profiles were progressively altered up to D5. The microbial composition profile then became unchanged from D5 to D7. This new information suggests that this in vitro model would be used to investigate the early biofilm formation and microbial behavior up to D5.
The authors didn’t mention how they selected the Ca concentration (why 1 and 5 mM? Please add some references). It would be great if they could have added higher concentration of Ca (10 mM and/or 50 mM) to their results. It has been reported that too much Calcium has a negative effect on bacterial attachment and biofilm forming.
RESPONSE: The rationale of Ca dose selection was explained in Results section in the original manuscript: Saliva has been reported to contain 0.6 mM~2 mM calcium ion [26,27] and dental plaque-released calcium concentration was shown to be 2 mM~5 mM [28]. Therefore, the higher Ca concentrations such as 10 mM or 50 mM are not relevant for our study investigating the effect of oral Ca environment.
The below mentioned references were not related to the study and hence re not included in the manuscript.
Chen G, Walker SL. 2007. Role of solution chemistry and ion valence on the adhesion kinetics of groundwater and marine bacteria. Langmuir. 23:7162–7169.
We agree with the authors here that the bacterial adhesion is governed by many factors, but the experiment set up is not relevant to our study.
Venegas SC, Palacios JM, Apella MC, Morando PJ, Blesa MA. 2006. Calcium modulates interactions between bacteria and hydroxyapatite. J Dent Res. 85:1124–1128.
This paper describes the diversity in bacterial behavior with different calcium concentrations and thus different from the oral community behavior.
Mangwani N, Shukla SK, Rao TS, Das S. 2014. Calcium-mediated modulation of Pseudomonas mendocina NR802 biofilm influences the phenanthrene degradation. Colloids Surf B Biointerfaces. 114:301–309).
This paper was not relevant to our study as it uses higher concentrations of Calcium.
It would be great if authors could add some more references for discussion part line 370-410.
The authors make many statements regarding that their results are in agreement with previous studies. These statements need to be revised. It has affected the novelty and uniqueness of their work. They should add some scientific arguments that support their claims.
RESPONSE: We revised the manuscript with this suggestion in mind. Thank you.
It would be great if authors could go through the article, and cover the typos, abbreviations, spacing error, and extra. The format should be constant all over the manuscript: Ex. 15ml or 15 ml (spacing error), day 1 or D1, 2 hrs or 2 hours (the format is not consistent), ul or µl, CaCl2 needs to be fixed (use subscript, CaCl2).
RESPONSE: These have been corrected.
Figures’ legend has different size and formatting. This needs to be fixed.
RESPONSE: To Editorial Office: Would you please comply with Reviewer 1’s suggestion? We do not have access to the manuscript formatting ability. Thank you.
Reviewer 2 Report
Dear author,
The topic of your study is interesting but for me, there is an important methodological problem.
L70-78 If I well understand you mixed the saliva of volunteers with undetermined periodontal status with either saliva of healthy volunteers or saliva of periodontal patients. I think it is a methodological problem and I don’t understand why? With this methodology, you don't have the group of healthy and the group of periodontal patients
Author Response
The topic of your study is interesting but for me, there is an important methodological problem.
RESPONSE: We appreciate your assessment on the significance and innovation of our study.
L70-78 If I well understand you mixed the saliva of volunteers with undetermined periodontal status with either saliva of healthy volunteers or saliva of periodontal patients. I think it is a methodological problem and I don’t understand why? With this methodology, you don't have the group of healthy and the group of periodontal patients
RESPONSE: We have reviewed the information of the periodontal status of all saliva donor subjects. The periodontal status revealed that our healthy donors did not have clinically diagnosed periodontitis. Their gingival sulcus/pocket depths were <4 mm. The manuscript has been revised to include the information.
Reviewer 3 Report
This is a very well-written article. Describe exactly what was performed and follow all the steps necessary for the research article writing.
It was my pleasure reading this article.
Congratulations to the authors.
Author Response
This is a very well-written article. Describe exactly what was performed and follow all the steps necessary for the research article writing.
It was my pleasure reading this article.
Congratulations to the authors.
RESPONSE: Thank you very much for the positive review and encouragement.
Reviewer 4 Report
The manuscript of “Effect of Calcium ion Supplementation on Oral Microbial Composition and Biofilm Formation in vitro” is reported that calcium affects biofilm formation. There are different calcium effects in Between health and periodontitis plaque microbiome in terms of formation biofilm by eDNA substrate. The prosses of Biofilm formation and following caries or periodontitis are major pathogen of oral disease, it is important the detection of the aggravation factors. Although this paper is of interest, several points are noted and are presented below.
Materials and methods:
How many samples did the author examined in this study?
The saliva samples were collected from undetermined periodontitis, healthy state (six person?) and periodontitis patients. But collected number of volunteers or patients were not described in materials and methods.
What means of “Undetermined periodontal status”?
In this study, oral microbiome is important factor, the microbiome is different between healthy, suffered from caries or periodontitis.
How treated the artificial oral microbials in following experiment?
The authors described that Oral microbials were treated from saliva stock derived from a pool. This pool was containing from healthy volunteers or periodontal patients? And also pool meanings how many parsons’ saliva was collected? Please indicate referenced paper that using saliva pool method.
Every data was not indicating the number of experimented samples. 16s rRNA gene sequencing was analyzed two samples in each group?
P4 L16: 2.816. S rRNA… means: 2.8. 16S rRNA ?
What was the purpose of the CFU analysis?
The samples were used 100µl aliquot of washed biofilm, which does not represent the number of bacteria in the biofilm. Dis the author want to determine the number of floating cells or the number of bacteria grown in the biofilm?
L358: Our results indicate that calcium enhanced biofilm biomass, increased metabolic and viability of oral microorganisms in a concentration-dependent manner.
What the reason that the colony counts are decreased overall groups?
It seems that if the culture medium is not changed for 5 days, the amount of eDNA and the bacterial flora will change because the nutrients will decrease and dead bacteria will increase.
While the addition of calcium to samples from the periodontal group increased diversity.
This suggests that the presence of calcium may have an effect on the number of colonies in the periodontally diseased group, but not Peptostreptococcus and Veillonella?
P11L42: The author describes that “This suggests that presence of calcium may lead to changes in eDNA, either through lysis of bacteria or through active release of eDNA in periodontitis.” If calcium affects bacterial fusion, I would think it would affect all bacteria, and the gene ratio of iDNA and eDNA similar result. How author express why do eDNA and iDNA have different percentages of bacteria?
Fig6: Periodontitis groups were seemed to show a greater change in flora with the addition of calcium; the group of Coccus and Veillonella are decreased, but what are the increased flora instead?
Author Response
The manuscript of “Effect of Calcium ion Supplementation on Oral Microbial Composition and Biofilm Formation in vitro” is reported that calcium affects biofilm formation. There are different calcium effects in Between health and periodontitis plaque microbiome in terms of formation biofilm by eDNA substrate. The prosses of Biofilm formation and following caries or periodontitis are major pathogen of oral disease, it is important the detection of the aggravation factors. Although this paper is of interest, several points are noted and are presented below.
RESPONSE: We agree with this reviewer on the significance and impact on the mechanism of oral microbial biofilm formation for the pathogenic development of oral diseases.
Materials and methods:
How many samples did the author examined in this study?
RESPONSE: In the initial study, we collected salivary samples from 6 healthy subjects and equally mixed to generate 1 experimental sample. In the validation study, we collected salivary samples from 2 healthy subjects and 2 patients with periodontitis. In the latter experiment, the salivary samples were not mixed and used as independent experiments.
The saliva samples were collected from undetermined periodontitis, healthy state (six person?) and periodontitis patients. But collected number of volunteers or patients were not described in materials and methods.
RESPONSE: The revised manuscript now defined the number of subjects.
What means of “Undetermined periodontal status”?
RESPONSE: This term was misleading and deleted. We have reviewed the information of saliva donor subjects. The periodontal status revealed that our healthy donors did not have clinically diagnosed periodontitis. Their gingival sulcus/pocket depths were <4 mm. The manuscript has been revised to include the information.
In this study, oral microbiome is important factor, the microbiome is different between healthy, suffered from caries or periodontitis.
RESPONSE: Our study showed the behavioral difference between oral microbial samples harvested from healthy subjects and patients with clinically diagnosed periodontitis.
How treated the artificial oral microbials in following experiment?
The authors described that Oral microbials were treated from saliva stock derived from a pool. This pool was containing from healthy volunteers or periodontal patients? And also pool meanings how many parsons’ saliva was collected? Please indicate referenced paper that using saliva pool method.
RESPONSE: In the initial study, salivary samples of 6 healthy volunteers were mixed at equal volume to generate 1 experimental sample. In the next validation study, we collected salivary samples from 2 healthy subjects and 2 patients with periodontitis. The salivary samples were not mixed and used as independent experiment.
Saliva pool method is well established and has been used for development of an oral biofilm in vitro. Please find the reference below:
Tian, Y., He, X., Torralba, M., Yooseph, S., Nelson, K., Lux, R., McLean, J., Yu, G. and Shi, W. (2010), Using DGGE profiling to develop a novel culture medium suitable for oral microbial communities. Molecular Oral Microbiology, 25: 357-367. https://doi.org/10.1111/j.2041-1014.2010.00585.x
Every data was not indicating the number of experimented samples. 16s rRNA gene sequencing was analyzed two samples in each group?
RESPONSE: Correct.
P4 L16: 2.816. S rRNA… means: 2.8. 16S rRNA ?
RESPONSE: This ‘Typo’ is corrected.
What was the purpose of the CFU analysis?
RESPONSE: We reviewed our study and concluded to delete the CFU analysis data from this manuscript.
The samples were used 100µl aliquot of washed biofilm, which does not represent the number of bacteria in the biofilm. Dis the author want to determine the number of floating cells or the number of bacteria grown in the biofilm? L358: Our results indicate that calcium enhanced biofilm biomass, increased metabolic and viability of oral microorganisms in a concentration-dependent manner. What the reason that the colony counts are decreased overall groups?
RESPONSE: The CFU measurement was conducted in a liquid culture condition, which would be suitable for a selected set of microbial species. However, other oral microbial species require solid culture conditions to accurately measure their CFU. Therefore, the reduction of CFU in our study can be induced by a variety of potential causes and may not accurately represent the overall microbial activity.
With this reviewer’s comment, we reanalyzed the validity of our CFU measurement and concluded that the CFU data were not relevant to the context and scope of our study. We deleted CFU data from the revised manuscript.
It seems that if the culture medium is not changed for 5 days, the amount of eDNA and the bacterial flora will change because the nutrients will decrease and dead bacteria will increase.
RESPONSE: The medium was not changed for all the conditions. Therefore, as pointed out, the reduction in nutrients may affect the long-term culture. In the revised manuscript, we present a new information including the long-term culture up to Day 7. As expected, there was a tendency to decrease biomass from D5 to D7, likely due to the decreased nutrients. However, the microbial composition examined by iDNA 16S rRNA PCR-DGGE did not show a major shift from D5 to D7. It must be noted that the change due to effect of calcium ions was more prominent.
While the addition of calcium to samples from the periodontal group increased diversity. This suggests that the presence of calcium may have an effect on the number of colonies in the periodontally diseased group, but not Peptostreptococcus and Veillonella?
RESPONSE: We agree with this reviewer’s interpretation of our data. We used the relative abundance of Peptostreptococcus and Veillonella as selected examples to demonstrate the shift in eDNA likely influenced by Ca ion supplementation.
P11L42: The author describes that “This suggests that presence of calcium may lead to changes in eDNA, either through lysis of bacteria or through active release of eDNA in periodontitis.” If calcium affects bacterial fusion, I would think it would affect all bacteria, and the gene ratio of iDNA and eDNA similar result. How author express why do eDNA and iDNA have different percentages of bacteria?
RESPONSE: The role of calcium ions to microbial viability and contribution to eDNA appears to be species specific. For example, S. mutans was reported to induce “suicide” mechanism under Ca supplementation condition through activating altA gene, leading to release of eDNA (Jung CJ, Hsu RB, Shun CT, Hsu CC, Chia JS. AtlA Mediates Extracellular DNA Release, Which Contributes to Streptococcus mutans Biofilm Formation in an Experimental Rat Model of Infective Endocarditis. Infect Immun. 2017 Aug 18;85(9):e00252-17. doi: 10.1128/IAI.00252-17. PMID: 28674029; PMCID: PMC5563578.).
The presence or activation of altA gene is not ubiquitous but rather limited to selected species. This is one example that Ca may affect selectively or more predominantly the selected species affecting the eDNA composition in the oral biofilm.
Fig6: Periodontitis groups were seemed to show a greater change in flora with the addition of calcium; the group of Coccus and Veillonella are decreased, but what are the increased flora instead?
RESPONSE: The authors would like to highlight that the addition of Calcium resulted in the decrease of the relative abundance of Peptostreptococcus while for Veillonella a significant increase in the relative abundance was observed,
Round 2
Reviewer 1 Report
The authors addressed most of the comments in the new version and the manuscript can be published.
Reviewer 2 Report
Dear authors,
thank you for considering my comments.